Full-annual demography and seasonal cycles in a resident vertebrate

Guimarães Murilo mu.guima@gmail.com 1
Correa Decio T. 2
Gaiarsa Marília Palumbo 3
Kéry Marc 4
1 Departamento de Zoologia, Universidade Federal do Rio Grande do Sul , Porto Alegre , Rio Grande do Sul , Brasil
2 Department of Integrative Biology, University of Texas , Austin , TX , United States of America
3 Department of Entomology, University of California , Riverside , CA , United States of America
4 Swiss Ornithological Institute , Sempach , Switzerland
Yoccoz Nigel
Electronic publication date: 2020 Feb 25
Publication date: 2020
Volume: 8
Electronic Location ID: e8658
Received 2019 Sep 18; Accepted 2020 Jan 29
Copyright: ©2020 Guimarães et al.
Copyright year: 2020
Copyright holder: Guimarães et al.
License: This is an open access article distributed under the terms of the Creative Commons Attribution License, which permits unrestricted use, distribution, reproduction and adaptation in any medium and for any purpose provided that it is properly attributed. For attribution, the original author(s), title, publication source (PeerJ) and either DOI or URL of the article must be cited.
License URL: https://creativecommons.org/licenses/by/4.0/

Keywords: Dynamics, Full-annual cycle, Mark-recapture, Periodicity, Population

Funding: CNPq 140684/20093 CAPES 2296/110 FAPESP 2008/54472-2 Ecology Graduate Program of UNICAMP Schifferli Scholarship Program of the Swiss Ornithological Institute This work was funded by CNPq (140684/20093), CAPES (2296/110), FAPESP (2008/54472-2), the Ecology Graduate Program of UNICAMP and the Schifferli Scholarship Program of the Swiss Ornithological Institute. The funders had no role in study design, data collection and analysis, decision to publish, or preparation of the manuscript.

==============================
Wildlife demography is typically studied at a single point in time within a year when species, often during the reproductive season, are more active and therefore easier to find. However, this provides only a low-resolution glimpse into demographic temporal patterns over time and may hamper a more complete understanding of the population dynamics of a species over the full annual cycle. The full annual cycle is often influenced by environmental seasonality, which induces a cyclic behavior in many species. However, cycles have rarely been explicitly included in models for demographic parameters, and most information on full annual cycle demography is restricted to migratory species. Here we used a high-resolution capture-recapture study of a resident tropical lizard to assess the full intra-annual demography and within-year periodicity in survival, temporary emigration and recapture probabilities. We found important variation over the annual cycle and up to 92% of the total monthly variation explained by cycles. Fine-scale demographic studies and assessments on the importance of cycles within parameters may be a powerful way to achieve a better understanding of population persistence over time.

Introduction

Wildlife demography encompasses the study of four vital rates that underlie numerical change in populations: birth, death, immigration and emigration (Conroy & Carroll, 2009). Understanding spatial and temporal patterns in these rates is essential for both ecology and management of living species; yet, we often lack sufficient resolution in our observations for a complete description of the demography of a species. For instance, studies on animal population dynamics usually focus on specific periods of the annual cycle (Doherty & Grubb, 2002; Singer et al., 1997), such as the breeding season, when a more sedentary lifestyle and increased activity increase the probability of detecting or catching individuals (Marra et al., 2015). This is the case for studies reporting individual variation in vital rates, such as survival probability (Grosbois et al., 2008), where individuals must be followed in the field over long periods (Lebreton et al., 1992). Although important, studies focusing on short periods typically can contribute less to our understanding of fundamental biological questions (Marra et al., 2015 and references therein).

Studies focusing on short periods or yearly effects may be useful to track population demography and dynamics over years. However, such studies may mask within-year variations, reducing the number of statistical points to document temporal variability and preventing identification of the most critical periods of the year (Grosbois et al., 2008). Recently, Marra et al. (2015) have emphasized the importance of events beyond the breeding season and argued in favor of demographic research during the full annual cycle to fully understand the population dynamics of a species.

Most full-annual-cycle research focuses on a small handful of key periods within and outside of the breeding season (Hostetler, Sillett & Marra, 2015), which for the vast majority of vertebrates extends for a period of one year, representing one time unit. Although more expensive and time consuming than assessments made only during a single period within a year, models describing events over the full annual cycle are useful both for theoretical and applied questions including carry-over effects (Harrison et al., 2011), and detection of population trends (reviewed in Hostetler, Sillett & Marra, 2015; Marra et al., 2015). Most demographic models considering the full annual cycle have been applied to long-distance migratory species (Hostetler, Sillett & Marra, 2015), especially birds (Culp et al., 2017; Rushing et al., 2017, but see Flockhart et al., 2015), although studies on resident species are also found for different taxa (Julliard et al., 1999; Martins, Guimarães & Verrastro, 2017; Da Rodrigues, Martins & Rodrigues, 2013). Resident species may offer better opportunities to study demography at a fine temporal scale, and may be subject to seasonal fluctuations due to less favorable periods for activity (Navas & Carvalho, 2010), even in tropical areas (Dingle & Drake, 2007).

Seasonal oscillations are expected for all living organisms (Panda, Hogenesch & Kay, 2002) and are intrinsically connected to variability in temporal predictors, such as photoperiod (Bradshaw & Holzapfel, 2007), moon phase (Hauenschild, 1960), temperature (Brown et al., 2004) and rainfall (Chaplin, 2001). Temporal predictors may be useful in revealing proximate or ultimate drivers of temporal variability, and are commonly employed to explain individual activity and population demography. However, temporal predictors are not always available or the relationship between such candidate predictors and temporal variability may not be clear (Crespin et al., 2002). Finally, temporal predictors may suffer from parameter estimability and are related to temporal variables, not to time itself (Fidino & Magle, 2017).

Within-year periodicity may often be fairly predictable, and accounting for such cyclicity in demographic rates may be better achieved by explicitly including cycles in population dynamics parameters. However, population dynamics models do not make extensive use of explicit within-season cycles and therefore, little is known about the magnitude of potential cyclic variation in population dynamics (Webster et al., 2002; Wingfield, 2008; Hostetler, Sillett & Marra, 2015; Marra et al., 2015). Here we explore the full annual cycle and demonstrate periodicity may be intrinsic in vital parameters. We investigate fine-scale, intra-annual demography using a resident vertebrate, the neotropical whiptail lizard, Ameivula ocellifera, through an intensive robust capture-recapture study (Pollock, 1982; Kendall et al., 1997; Rankin et al., 2016). We first describe the temporal variation throughout the annual cycle, and then, we evaluate the presence and magnitude of cyclic effects in demographic parameters of this resident neotropical species.

Methods

Study site and species

We conducted the study in the Estação Ecológica de Jataí (21°37′3″S, 47°45′55″W), a transitional area between the Cerrado and the Atlantic Forest biomes in the state of São Paulo, SE Brazil. Mean temperatures vary between 11 °C and 30 °C in the coldest and hottest months, respectively. Annual rainfall is around 1,500 mm, mostly concentrated in the rainy season, from October to March (CEPAGRI, 2011).

We surveyed a population of the whiptail lizard, Ameivula ocellifera, a small (30–78 mm Snout-vent length), and heliophilous Teiidae lizard that occurs in most parts of tropical Brazil (Mesquita & Colli, 2003). Individuals are active year-round, especially in the rainy season (austral spring-summer) when breeding occurs. Newborns are found in the austral summer, from January to March (Guimarães et al., 2017). More details about the species may be found in Mesquita & Colli (2003).

Sampling design

We sampled the population using a regular, square trapping grid consisting of 121 pitfall traps 25 m apart, over 6.2 ha, from September 2010 to September 2011, capturing individuals for seven consecutive days in each of 13 consecutive months. We used digital photography and batch marking, by clipping the third joint of the second toe of the right hand (ICMBio Animal Welfare Permit 10423-1) to recognize individuals. On each capture, we determined sex and assigned individuals to three sex/age classes: adult males (SVL > 40 mm), adult females (SVL > 51 mm), and newborns (SVL < 37 mm; non-reproductive individuals in their first year, characterized by the presence of umbilical scar). After capturing and marking, we released individuals near the same trap where they had been caught. We used the Interactive Identification System software (Van Tienhoven et al., 2007) to identify individuals based on color and scale patterns. For more details, see Guimarães et al. (2017).

Statistical analysis

We modeled our mark-recapture data with the full-capture hierarchical robust design (RD) model (Rankin et al., 2016) and used a Bayesian mode of inference with MCMC techniques, to assess the effects of temporal variation on demographic parameters. The RD model distinguishes primary periods (here a 7-day period), between which a population is assumed to be open, and nested secondary periods (here, daily capture occasions), during which the population is assumed to be closed.

We distinguished three demographic groups in our analyses: adult males, adult females, newborns. We expected different demographic rates for them and therefore, in our models, we stratified all parameters by these groups. In our analysis, we focused on three main biological parameters in the models: apparent survival probability (φ, hereafter ‘survival’), which is the product of true survival and site fidelity, and thus, permanent emigration and death are not distinguished; temporary emigration probability, the probability of temporarily leaving the study site given the individual was onsite (γ″, hereafter ‘emigration’), and detection probability (p, hereafter, ‘recapture’), the probability of recapturing an individual given it is onsite. We also estimated abundances for group and month using parameter-expanded data augmentation (PX-DA, Royle & Dorazio, 2012), which we do not focus on here but describe in the Appendix S1. Our model also provides the probability of staying off-site (γ′), given an individual was already off-site in the previous sampling occasion, and the probability of entering the study site over time (pent). The parameter γ′ is hard to estimate and pent is really more a nuisance parameter than of real biological interest (Rankin et al., 2016). Therefore, we do not discuss them in any detail.

We fit two variants of the RD model to our data that only differ in terms of the specification of the time variation in the parameters. In model 1, time was included as a random effect in the three parameters survival, emigration and recapture. That is, in this model we allowed parameters to be different for each month and to vary around a constant value according to a random effect. In model 2, we added cyclic effects of time on the same three parameters, survival, emigration, and recapture, using a periodic trigonometric function of month, and on top of that specified monthly residuals, also as random effect (see also Flury & Levri, 1999; Crespin et al., 2002). We added two effects of month into the linear predictor of these three parameters, as follows: alpha1∗ cos2∗pi∗monthT+alpha2∗sin2∗pi∗monthT where pi is the number Pi, month corresponds to the month number between 1 (January) and 12 (December), T represents the length of the periodic response (here T = 12), and alpha1 and alpha2 are the two coefficients associated with the covariate ’month’. In this formulation, month/T tells us how far through the full annual cycle of 2*pi each month is (see model code in the Appendix S2). In model 2, the proportional reduction in the among-month variation achieved by specification of the cyclic patterns in these parameters enabled us to quantify the proportion of the among-month variation that could be explained by the cycles (Kéry & Schaub, 2012). Both models were built together, in an a priori comparison.

We used vague priors and hyperpriors for all parameters, which we show in the Online Appendix S2. We fit our models in the BUGS language (Lunn et al., 2000) using program JAGS (Plummer, 2003), run from R (R Core Team, 2018) through the jagsUI interface (Kellner, 2014). We ran three chains with 400,000 iterations each, discarding 300,000 as burn-in and thinning at a rate of 1 in 100. We determined chain convergence by visual examination of trace plots and by the Brooks–Gelman–Rubin statistic (Brooks & Gelman, 1998), which was <1.1 for all parameters. We present posterior means and 95% credible intervals (CRI).

Results

Our field effort consisted of 11,011 trap-days (121 traps * 7 days per month * 13 months) during which we captured 164 adult males, 163 adult females and 89 newborns for a total of 416 individuals. Fifty-one adult males, 29 adult females and 14 newborns were recaptured at least once.

Under model 1 (with random month effects), mean monthly apparent survival probability was higher for adult males (0.94, CRI [0.81–0.99]) and adult females (0.90, CRI [0.72–0.99]) than for newborns (0.64, CRI [0.17–0.96]). We observed considerable variability in survival during the course of a year, where adults of both sexes presented similar patterns through the year, and newborns showed a sharp decrease after hatchling, in the month of March (Fig. 1). Under model 1, mean annual survival during our study was 0.17 (0.01–0.39) for males, 0.18 (0.01–0.60) for females and 0.002 (5.62e-12–0.02) for newborns.

Figure 1 Monthly probabilities of survival, emigration and recapture for adult males, adult females and newborns Ameivula ocellifera, between September 2010 and September 2011.

(A–C) Model based on random month effects only (model 1). (D–F) Model based on cycles plus random monthly deviations from these cycles (model 2). Vertical gray lines (A, B, C) and shaded areas (D, E, F) represent 95% credible intervals.

The newborns showed the highest probability of emigration (0.32, CRI [0.01–0.85]), followed by adult females (0.16, CRI [0.03–0.63]) and males (0.15, CRI [0.042–0.66], Fig. 1). Daily mean recapture probabilities were similar in all three groups with small variations during the year, where males and newborns presented the highest rates during the breeding season, at the beginning and in the end, respectively (Fig. 1).

Model 2 (with cycles added) revealed a strong periodicity in all three parameters that we modeled with a cosine periodic function of the month (survival, emigration and recapture; Fig. 1). Comparisons of the magnitude of the (residual) among-month variance between models 2 and 1 showed that annual cycles explained 92.5% of the monthly variation in apparent survival, 86.8% in emigration probability, and 84.6% in recapture probability.

Discussion

Many population dynamics studies are based on the observation of a population just once a year, often during the breeding season. Recently, it has been pointed out that studies with high temporal resolution are required to fully understand population dynamics, so are models covering the full annual cycle of a species, i.e., with multiple samplings a year (Hostetler, Sillett & Marra, 2015; Marra et al., 2015). Here, we modeled fine-scale temporal variability in the vital rates of a resident species and assessed the population throughout the year, including low-activity periods. We showed that demographic parameters varied along the full annual cycle in a cyclic fashion, suggesting that the strong periodic signal found in those vital parameters could be associated to long-term demography. Our findings provide detailed information on the population biology of a species and were only possible due to the high-temporal resolution of demographic information for an entire annual cycle.

The full annual cycle reveals detailed information on species biology

Survival estimates for the vast majority of the ∼6,500 species of lizards worldwide are lacking, but general patterns have emerged based on life history traits, including foraging mode and mating system. Our results contribute to the relationship between clutch size and growth rate—the classical slow-fast continuum—, which finds support in lizard natural history where early investments in rapid growth may decrease late survival (Clobert, Garland Jr & Barbault, 1998; Olsson & Shine, 2002). Survival in slow-growing, territorial lizards from the Iguania clade are said to be higher (Ujvari et al., 2015; Iverson et al., 2016; Keehn, Shoemaker & Feldman, 2019), than in the non-territorial Scleroglossa clade, to which the Teiidae family belongs (Pianka & Vitt, 2003). Males from Teiidae species can present female accompaniment (Pianka & Vitt, 2003) and fight conspecific males to guard females, which may impair survival (Ancona, Drummond & Zaldívar-Rae, 2010). Females in turn, may also pay a high survival cost in the breeding season due to basking activity and low mobility (Shine, 1980). After the breeding season, we found that survival dropped 30% for adult males and 35% for adult females. Thus, carry over effects due to the costs of reproduction may affect future short-term and long-term survival. Although the duration of our study prevents us from linking the observed increased mortality to costs of reproduction, the full annual cycle research may be promising to uncover sources of mortality in species.

Temporary emigration for adults was the highest after the mating and hatchling periods, coinciding with the beginning of the dry and cold season. Energetic reserves may be depleted after the mating season (Shine, 1980) and we found several individuals buried 10–20 cm below ground in our study site during this period of the annual cycle. Additionally, most individuals recaptured in our study were found up to 50 m from the trap they were first caught and similar home ranges were already reported for other Teiidae lizards (Hernández-Gallegos et al., 2018; Winck, Blanco & Cechin, 2011). Considering that site fidelity may be high in the studied species, vertical migration could explain temporary emigration. Temporary emigration probability of newborns was up to four times higher than that of adults, explaining, at least in part, the scarcity of juvenile demographic information that is common across taxonomic groups, because most juvenile individuals are just unavailable for capture. This pattern is usually attributed to a variety of aspects, including smaller body sizes, secretive and inconspicuous habits, and high mortality rates (Rivas et al., 2016; Bailey et al., 2017; Wilson et al., 2018).

Adult males presented the highest recapture rates at the onset of the breeding season, coinciding with high mating activity rates during this period. Since our trap system was stationary, we expect capture rates to increase with activity, and in this case, detection probability parameter of a capture-recapture model is not merely a nuisance parameter. More than that, detection probability is a parameter with a biological meaning, related to physical activity, which produces a strong signal in the detection probability of species (Strebel et al., 2014; Sutherland et al., 2016). However, detection probability may be also related to other characteristics, such as capture effort and trapping methods. In territorial species, males may present higher recapture rates than females due to greater site fidelity. Overall, our mean recapture estimates were similar across groups. Given that this species feeds primarily on termites, where all individuals actively forage in open areas (Mesquita & Colli, 2003), our results may be related to their feeding behavior and the absence of territoriality (Pianka & Vitt, 2003).

Periodicity can be strong in demographic rates

Cyclicity is found in endogenous rhythms of every life form, from cells to individuals, and the biological clock is influenced by exogenous rhythms, such as Earth’s rotation and climate, to maximize fitness (Panda, Hogenesch & Kay, 2002; Wingfield, 2008). This explains, at least in part, why activity in many organisms is concentrated in short time frames, such as breeding events (Bradshaw & Holzapfel, 2007; Harrison et al., 2011; Marra et al., 2015). Although the full annual cycle is mostly described for long-distance migratory species, where individuals are unobservable for some time, tropical species may also be subject to periodicity due to temporary absences, involving reduced activity and unavailability.

Surprisingly, few studies exploring the life cycle of a species have actually fitted cyclic patterns in demographic rates. If we assume that the beginning of a temporal pattern in a parameter to “meet” with the end of it over the course of a year, such cycles are almost to be expected. Periodicity may occur at different time scales, such as observed in the daily activity of the New Zeland mudsnails (Flury & Levri, 1999), or in the yearly activity of American mesocarnivore mammals, with up to 75% of temporal variability in colonization rates explained by cyclicity (Fidino & Magle, 2017). We fitted cycles in three demographic parameters (apparent survival, emigration probability and recapture rate) and found strong evidence of annual periodicity, with cycles explaining on average a remarkable 88% of the total variation among months. We are aware that the observed variability in the demographic parameters was based on only one full cycle, preventing us from claiming that the patterns found here represents long-term variability. However, the monthly variation we found supports the cyclicity of life forms suggesting that the within-year pattern observed may be seen among years as well. Detailed, multi-year information on population demography may support our findings on the magnitude of specie’s annual cycles, which could be of great use to wildlife management and conservation.

In the face of rapid biodiversity declines, information on individuals and populations are essential for conservation purposes. Between-year and within-year population assessments may uncover different aspects of population demography and dynamics. For example, bank vole populations were manly driven by the acorn mast crop production from year to year, whereas within-year variation was mainly attributed to emigration, a population intrinsic factor (Crespin et al., 2002). Between-year estimates may be useful to assess population growth over years while within-year population assessments may reveal the most sensitive periods of the full annual cycle. In this way, full annual cyclic modeling, the combination of full annual cycle research with cyclic parameter estimation, can be powerful to uncover critical points in the trajectory of populations, supporting decision-making actions.

Conclusions

We endorse the use of full annual cycle studies of species, and suggest that investigations on resident species may provide cheaper fine-scale information than long-distant migratory species. With our results, we raise the possibility that intra-annual temporal variability in demographic rates may be highly cyclic. Periodicity should be more explored in vital rate assessments, and here we claim for its inclusion in population demography as a venue for predictive studies. Fluctuations may be triggered by many different sources, and understanding the mechanisms that regulate populations over time is of fundamental importance to ensure the persistence of species. Full annual cyclic modeling may be important to predict future population oscillations and raise important insights, even when they explain little variability.

Supplemental Information

Appendix S1 Abundance estimation procedure

Click here for additional data file.

Appendix S2 R code and JAGS model

Click here for additional data file.

Data S1 Mark-recapture dataset

The archive has the format .inp, which is the standard to enters R through R package RMark.

Click here for additional data file.

We thank Robert Rankin and Brett McClintock for advice on statistical analysis. Ricardo Sawaya, Hamanda B. Cavalheri, Sérgio Serrano, Thiago Oliveira, Roberto Munguia Steyer, Glauco Machado and our field crew were especially kind and helpful during data collection. We also thank Jeffrey Hostetler, Nigel Yoccoz and an anonymous reviewer for valuable suggestions that improved our paper. The ICMBio provided the permission for data collection.

Additional Information and Declarations

Competing Interests

Author Contributions

Animal Ethics

Data Availability

The authors declare there are no competing interests.

Murilo Guimarães conceived and designed the experiments, performed the experiments, analyzed the data, prepared figures and/or tables, authored or reviewed drafts of the paper, and approved the final draft.

Decio T. Correa and Marília Palumbo Gaiarsa performed the experiments, authored or reviewed drafts of the paper, and approved the final draft.

Marc Kéry conceived and designed the experiments, analyzed the data, authored or reviewed drafts of the paper, and approved the final draft.

The following information was supplied relating to ethical approvals (i.e., approving body and any reference numbers):

Instituto Chico Mendes de Conservação da Biodiversidade provided a permit for this research (License 10423-1).

The following information was supplied regarding data availability:

Raw data and code are available as Supplemental Files.

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
