# Peer review of "Full-annual demography and seasonal cycles in a resident vertebrate"

_PeerJ, doi:10.7717/peerj.8658_

## Round 0.1 · original submission · Major Revisions

I like the paper and I concur with the reviewers that it is an interesting and robust study. It is also correct that you should better explain how the within-year approach will help us understanding better population dynamics (as compared to having only yearly estimates - see a recent example in Chiffard et al. 2019). I would also like to point out that a similar approach to estimating within-year survival - and assessing both within and between year variation (with sin/cos functions for the within year cycle, ie Fourier series) was used in Crespin et al. (2002).

Chiffard, J., A. Delestrade, N. G. Yoccoz, A. Loison, and A. Besnard. 2019. Warm temperatures during cold season can negatively affect adult survival in an alpine bird. Ecology and Evolution. 10.1002/ece3.5715

Crespin, L., R. Verhagen, N. C. Stenseth, N. G. Yoccoz, A. C. Prévot-Julliard, and J.-D. Lebreton. 2002. Survival in fluctuating bank vole populations: seasonal and yearly variations. Oikos 98:467-479.

Reviewer 1 ·

Basic reporting

The authors demonstrate how cyclic variability can describe variations in monthly demographic rates for a resident tropical lizard. The objective of identifying the advantages to full cycle demographic rate estimation is useful, however the authors overstate the novelty of this approach. The authors examine within year variability of survival and migration rates. It is relatively common to examine variations of within year survival rates for many species of mammals, birds, and reptiles, particularly for non-migratory species (game species in particular provide a wealth of literature on this). That is not to say the objective is not worthwhile, but the authors repeatedly mention how very few studies examine this and that is not correct.
While there are many studies that do examine within year variations in demographic rates, the authors are correct that it is more common to examine demographic rates during a single season (usually the breeding season). These single season assessments are useful in tracking population changes and examining how annual fluctuations in climate or habitat, etc. relate to these demographic estimates. Since the authors are advocating for more research on the full cycle demographic variability, it would be useful to put the introduction and discussion into the context of when the different types of estimates should be used, and when a single season approach may be insufficient. What specifically can we do with a full cycle demographic data that we could not do with a single season estimate? When should we bother with the full season? How would management or conservation change with that information?
If the introduction and discussion were restructured to emphasize the information gained from a full cycle approach and the implications of that information for conservation or ecology, then the discussion the species biology could be more easily incorporated into the study objectives. Currently, the species specific information is disjointed with the rest of the paper. But if comparisons are made back to how this additional information can be useful (and is not obtained in single season demographic assessments) it would help tie the paper together.

Experimental design

Line 144: What is the error rate on identification for this method?
The robust design approach used assumes closure within a month and allows for instantaneous transitions between months. Describe the possible implications for this assumption and how might the results be impacted if transitions are not instantaneous. The approach is acceptable, but you should recognize how the interpretations are impacted by this design.

Validity of the findings

Lines 203-206: The logic here was confusing. Why did the survival drop after reproduction when the previous sentences suggest survival might be lower during reproduction?
Lines 212-214: Why is the high newborn emigration probability explaining the scarcity of juvenile demographic information?
Line 221: Authors should discuss other reasons where capture rate might increase that is not related to activity rates. Is there any reference that supports this? Could the capture rates not vary with resource availability or predator levels that are not directly correlated with activity only?
Lines 251-253: This sentence is great and finally helps me see your argument for examining full cycle demography. Bring information like this to the introduction and expand on it here!

Additional comments

Line 26 (and line 47). Specify wildlife demography or similar, human demography generally refers to different criteria.
Line 50: Authors state we often lack sufficient resolution in our observation. But lack the resolution to do what?
Line 63: “represents annual events” is unclear.
Line 77: Give examples of temporal predictors.
Line 78: Avoid using ‘they’, what are you referring to here?
Line 160: Explain how 11,011 trap-days is achieved. How many days vs how many traps.
Line 190: Be more explicit than “broaden the understanding”. I’m still unclear on the benefit of the study from this sentence.
Line 194: The ~6500 species of lizard worldwide?

·

Basic reporting

For the most part, the writing and other reporting are clear and well done.
Lines 73 – 75: I would add temperature and rainfall to the list of possible temporal drivers.
Lines 75 – 76: The paragraph this sentence is in is a bit confusing. If I’m reading it correctly, the authors could make it clearer in this sentence by confirming that “temporal drivers” and “temporal predictors” are referring to the same things in somewhat different contexts.
Study species section should make clear when hatching occurs and when newborns become adults. Can individuals transition from newborn to adult within the study year?
Line 190: Specie’s population biology should be species’ population biology, I think.
204: “but may present survival decrease due to basking activity…” should be rephrased. Something like “but may have lower survival due to basking activity…” might work.
205: “30% to 35%” should be “30% or 35%,” I think.
Line 238: See comment on line 190.
Figure 1 is compact and information-rich (only figure), and that is admirable. But it’s a bit tricky to make out many things. The symbols should be larger or in some other way clearer, as it’s hard to tell them apart. The heading should designate the period of study, in part to make the x-axis a little clearer. If I understand the figure correctly, the results from model 2 show neither uncertainty (credible intervals) around the cycles nor the monthly random effects. At least the first of these (preferably both) should be included, especially given the high uncertainty shown around the model 1 results. These changes may require splitting the results up into more panels or more figures.

Experimental design

In general, the experimental design seems sound and well described.
Lines 73 – 90: This is a key paragraph for the justification of this approach. It is true that data on relevant “temporal predictors” are often not available. Temporal drivers offer more than understanding the importance of temporal variability, however. Ideally, they indicate the proximate or ultimate drivers of it. As such, they provide a somewhat more mechanistic approach to understanding temporal variability in demography than the cyclic approach worked out here.
Line 100: May want to define SVL
Lines 109-111: Section on determining class by length doesn’t make a lot of sense. Are sizes never in between?
Lines 124-129: May want to make clear apparent survival opposite of permanent emigration, not temporary
Lines 131-133: Why is pent a nuisance parameter? Isn’t that recruitment? I think the case may be different in the Rankin paper, where dolphins must acquire marks (naturally?) before they can be entered into the CMR dataset. In addition, there was informal language in this sentence.
Line 167: This would be clearer to me if the species information had spelled out the annual cycle of this species a bit more. When in the year is the hatchling phase?

Validity of the findings

In general, the approach and the findings seem valid. However, I have one set of serious concerns, all related to the strength of inference that can be made about full-annual-cycle (FAC) population dynamics from this study.
1) I can’t speak for my coauthors or others that have written about FAC population modeling, but for me, a key component is modeling *events* that happen within the year, or in some sense capturing dynamics with consequences beyond the year. Why do cycles in survival, emigration, or recapture probabilities within the year matter for long-term population dynamics? What would a population model parameterized with annual demographic rates get wrong in this case (assuming the population census was timed correctly within the population model)? Some speculation might be required here, of course.
2) In general, there is a lot of inference made from one year of data (see specific comments below).
3) FAC population modeling is about modeling both within and between years. This manuscript is about demographic estimation, not population modeling, of course. But understanding population dynamics requires understanding year-to-year variation in demography at least as much as it requires understanding within year variation in demography.
This is not to say I think this study is useless or that it requires more years of data to be published. I believe this study is valuable and the approach used is interesting and potentially useful for other studies. But I think some of the claims in the introduction and discussion about what was accomplished need to be tempered.
Was Model 2 designed before or after looking at variation in vital rates (e.g., through running Model 1)? In other words, was it developed a priori or post-hoc? Either is fine, but it’s good to be explicit, and with only one year of data, the answer affects the strength of inference for cycles in general.
Statement in lines 188 – 190 may need more than one year’s data to justify.
Lines 206 – 208: This sentence goes too far in a few ways, I think. Although carryover effects have been demonstrated in many other species, the authors have not proposed a mechanism for this one or explained why the survival cost of breeding would come a few months later. There may be many other things that differed over that period compared to the rest of the (or that) year – there’s little evidence presented that the drop in survival probability was due to reproduction.
Lines 209 – 210: Why would lower energetic reserves lead to temporary emigration?
Lines 210 – 212: This explanation makes more sense to me. I think it should be elaborated on a bit.
Lines 246 – 248: Again, what’s the justification for this claim? Why should the reader believe that the pattern over one year represents something more general?
Lines 253 – 256: I’m not sure this is a point specifically illustrated by the results.
Lines 256 – 257: Not demonstrated with only one year of data.

---

## Round 0.2 · accepted · Accept

This is a field I know well (the authors used an approach which is similar to one developed in a paper from 2002 that I co-authored). You have made a thorough revision of the paper, answering the constructive comments of both reviewers.

I enjoyed reading this new version of the paper and hope it will lead to further work.